# Maximum Margin Based Activation Clipping for Post-Training Overfitting Mitigation in DNN Classifiers

## Abstract

Well-known (non-malicious) sources of overfitting in deep neural net (DNN) classifiers include: i) large class imbalances; ii) insufficient training set diversity; and iii) over-training. In recent work, it was shown that backdoor data-poisoning *also* induces overfitting, with unusually large classification margins to the attacker's target class, mediated particularly by (unbounded) ReLU activations that allow large signals to propagate in the DNN. Thus, an effective post-training (with no knowledge of the training set or training process) mitigation approach against backdoors was proposed, leveraging a small clean set, based on bounding neural activations. Improving upon that work, we threshold activations *specifically to limit maximum margins (MMs)*, which yields performance gains in backdoor mitigation. We also provide some analytical support for this mitigation approach. Most importantly, we show that post-training MM-based regularization substantially mitigates *non-malicious* overfitting due to class imbalances and overtraining. Thus, unlike adversarial training, which provides some resilience against attacks but which *harms* clean (attack-free) generalization, we demonstrate an approach originating from adversarial learning that *helps* clean generalization accuracy. Experiments on CIFAR-10 and CIFAR-100, in comparison with peer methods, demonstrate strong performance of our methods.

## 1 Introduction

Deep Neural Networks (DNNs), informed by large training datasets and enhanced by advanced training strategies, have achieved great success in numerous application domains. However, it may be practically difficult and/or expensive to obtain a sufficiently rich training set which is fully representative of the given classification domain. Moreover, some classes may be very common, with other classes quite rare. Large-scale datasets, with many classes, often have a long-tailed label distribution (Buda et al., 2018; Liu et al., 2019). Moreover, for tasks such as fraud detection (Bolton & Hand, 2002) and rare disease diagnosis (Kononenko, 2001), it is difficult to obtain sufficient positive examples. Thus, high class-imbalance may exist in the training set, in practice. Also, in Federated Learning (Li et al., 2023) class imbalance is a serious issue because the training data is distributed and private to each client. Thus, class imbalances could be heterogeneous across clients and *unknown* even to the centralized server (Wang et al., 2021). Classifiers trained on unbalanced data will tend to bias predictions toward (overfit to) the common classes, with poor accuracy on test samples from rare categories. Classifier overfitting may also result from over-training or from insufficient training set diversity, irrespective of class imbalance.

Recently (Wang et al., 2022), it was shown that backdoor poisoning of the training dataset *also* produces a (malicious) overfitting phenomenon, with unusually large maximum classification margins for the attacker's designated target class, relative to the margins for other classes. Such overfitting is particularly aided by (unbounded) ReLU activations[1], which allow large signals to propagate in the DNN. This enables samples containing the backdoor trigger to "defeat" the features that would otherwise induce classification to the class of origin of the sample (its "source" class), causing the DNN to instead decide to the attacker's target class.

---

[1]ReLUs are often used to accelerate learning by ensuring the associated gradients have large magnitudes.

## 2 RELATED WORK

To address overfitting due to class imbalance, numerous *during training* methods have been proposed. *Data balancing* methods resample the training set to balance the number of training examples across classes (Chawla et al., 2002; Drummond et al., 2003; Shen et al., 2016). *Loss reweighting* methods give greater weight to samples from rare categories when computing the training loss (Cao et al., 2019; Cui et al., 2019; Zhang et al., 2021). However, the above two methods are ineffective at mitigating overfitting in the case of extreme class imbalance. *Transfer Learning* methods (Liu et al., 2018b; Yin et al., 2019) transfer knowledge learned from the common classes to the rare classes. However, these methods substantially increase the number of model parameters and learning complexity/training time.

There are also some methods proposed to address the over-training problem (Ying, 2019). One of the simplest methods is early stopping (Raskutti et al., 2014) based, e.g., on classification accuracy evaluated on a validation set. Other methods aim to mitigate the impact of over-training by pruning (Blalock et al., 2020) or regularizing (Girosi et al., 1995; Srivastava et al., 2014) the network.

(Wang et al., 2022) mitigated overfitting associated with backdoor data poisoning by imposing saturation levels (bounds/clipping) on the ReLU activation functions in the network. Their approach, called Maximum-Margin Backdoor Mitigation (MM-BM), leveraged a small clean dataset and was applied *post-training*, without requiring any knowledge of the training set (including the degree of class imbalance, i.e. the number of training samples from each class) or of the training process (e.g., how many training iterations were performed, and whether or not a regularizer was used during training). Such information is not available in practice if the defense is applied, e.g., by an entity that purchases a classifier from a company (e.g., a government, or app classifiers downloaded onto cell phones). It is also unavailable in the *legacy classifier* scenario, where it has long been forgotten what training data and training process were used for building the model. MM-BM does not even require knowledge of whether or not there was backdoor poisoning – it substantially reduces the attack success rate if the DNN was trained on backdoor-poisoned data, while only modestly reducing generalization accuracy in the absence of backdoor poisoning (and on clean test samples even if there was backdoor poisoning). Finally, MM-BM does not rely on any knowledge of the mechanism for incorporating the backdoor trigger in an image (the backdoor embedding function) – MM-BM is largely effective against a variety of backdoor attacks.

By contrast, other backdoor mitigation methods, such as NC mitigation (Wang et al., 2019) and I-BAU (Zeng et al., 2021), are only applied if a backdoor is first detected, rely on accurate reverse-engineering of the backdoor trigger, and often assume a particular backdoor embedding function was used (NC assumes a patch-replacement embedding function). If the attacker's embedding function does not match that assumed by the reverse-engineering based detector, the detector may fail. Moreover, even if the detector succeeds, the reverse-engineered backdoor trigger may not closely resemble that used by the attacker – in such cases, these mitigation methods, which rely on accurate estimation of the backdoor trigger, may fail to mitigate the backdoor's operation. Other mitigation methods also have limitations. Some methods seek to perform data cleansing prior to training the (final) classifier, e.g., (Chen et al., 2018; Tran et al., 2018; Xiang et al., 2021; Wang et al., 2023). These methods require access to the training set. Moreover, there is Fine-Pruning (Liu et al., 2018a), which removes neurons deemed to not be used by (to activate on) legitimate samples; but this approach is largely ineffective at backdoor mitigation.

(Wang et al., 2022) proposed both a backdoor detector (MM-BD) and a mitigator (MM-BM). The detection statistic used was the maximum margin (MM), with detections made when the estimated maximum margin (over the feasible input domain) for one class is unusually large relative to that for other classes. However, somewhat surprisingly, MM-BM was not based on minimizing (or penalizing) the maximum margins produced by the network – the objective function they minimized, in choosing neural saturation levels, has one term that encourages good classification accuracy on the available clean set, with the second term penalizing the sum of the norms of the saturation levels (with a saturation level parameter for each neuron (or each feature map, for convolutional layers)). Since they characterize overfitting by unusually large MMs, it seems natural to instead *mitigate* overfitting by explicitly *suppressing* these MMs. Moreover, while (Wang et al., 2022) recognized backdoors as an overfitting phenomenon, they did not investigate whether their approach might

mitigate overfitting attributable to other causes (non-malicious overfitting, e.g., associated with class imbalance).

In this paper, we address these limitations of MM-BM. That is, we propose post-training maximum classification-margin based activation clipping methods, ones which *explicitly* suppress MMs, with these approaches respectively addressing malicious overfitting and also non-malicious overfitting (e.g., class imbalance). Our MM-based strategies outperform MM-BM and other peer methods for backdoor mitigation (while only modestly affecting accuracy on clean test samples) and substantially improve generalization accuracy in the non-malicious case (class imbalances and over-training). Moreover, our approaches require no knowledge of the training set or knowledge and/or control of the training process, unlike methods discussed earlier. Our methods are also relatively light, computationally.

To the best of our knowledge, ours is the first work to address class imbalances post-training. Finally, this paper demonstrates that backdoor defense ideas from adversarial learning, which ostensibly address a security threat, also have more general applicability in *improving* the robustness of deep learning solutions. This in contrast, e.g., to adversarial training (Madry et al., 2018) which provides some benefit against adversarial inputs (test-time evasion attacks) but may significantly degrade accuracy on clean test data (in the absence of attacks).

## 3   MAXIMUM MARGIN BASED ACTIVATION CLIPPING

Suppose that the DNN classifier $f(\cdot)$ has already been trained and that the network weights and architecture are known. Let $\mathcal{D}$ be a small, balanced dataset available for purpose of overfitting mitigation ($\mathcal{D}$ is wholly insufficient for retraining the DNN from scratch). Let $h_\ell(\cdot)$ be the activation-vector of the $\ell^{\text{th}}$ (feedforward) layer. When given an input example $\mathbf{x}$, the output of an $L$-layer feedforward network can be expressed as:

$$f(\mathbf{x}) = h_L(\cdots h_2(h_1(\mathbf{x}))).  \quad (1)$$

That is, $h_\ell(\cdot)$ represents an internal layer (e.g., convolutional layer, linear layer), along with the corresponding pooling operation (for convolutional layers), and batch normalization.

We aim to adjust the decision boundary of the network by introducing a set of bounding vectors $\mathbf{Z} = \{\mathbf{z}_\ell, \ell = 1, ..., L - 1\}$ to limit the internal layer activations, where $\mathbf{z}_\ell$ is a vector with one bounding element for each neuron in layer $\ell$. For DNNs that use ReLUs (as considered herein), which are lower-bounded by zero, we only need to consider upper-bounding each internal layer $h_\ell(\cdot)$, with the bounded layer represented as[2]:

$$\bar{h}_\ell(\cdot; \mathbf{z_1}) = \min\{h_\ell(\cdot), \mathbf{z}_\ell\},  \quad (2)$$

and with the overall bounded network expressed as:

$$\bar{f}(\mathbf{x}; \mathbf{Z}) = h_L(\bar{h}_{L-1}(\cdots \bar{h}_1(\mathbf{x}; \mathbf{z}_1) \ldots; \mathbf{z}_{L-1})).  \quad (3)$$

Notably, for convolutional layers, since the output feature maps are spatially invariant, we can apply the same bound to every neuron in a given feature map. So the number of elements in the bounding vector of a convolutional layer will be equal to the number of feature maps (channels), not the number of neurons.

### 3.1   MITIGATING BACKDOOR ATTACKS: MMAC AND MMDF

Let $\mathcal{D}_c \subset \mathcal{D}$ be the clean samples originating from (and labeled as) class $c \in \mathcal{Y}$. Also for each class $c \in \mathcal{Y}$, let $f_c(\cdot)$ be the class-$c$ logit (in layer $L - 1$), and let $J_c$ be the number of different estimates of maximum classification margin starting from different randomly chosen initial points ($J_c > 1$ owing to the non-convexity of the MM objective and to the potential need to "tamp down" not just the global maximum margin but also large margins (associated with overfitting) that may occur anywhere in the feasible input space). Thus, to mitigate the backdoor in a poisoned model,

---

[2]This is a vector 'min' operation, i.e. with neuron $k$'s activation $h_{lk}(\cdot)$ clipped at $z_{lk}$.

the following Maximum-Margin Activation Clipping (MMAC) objective is minimized:

$$L(\mathbf{Z}, \lambda; \mathcal{D}) = \frac{1}{|\mathcal{D}| \times |\mathcal{Y}|} \sum_{\mathbf{x} \in \mathcal{D}} \sum_{c \in \mathcal{Y}} [\bar{f}_c(\mathbf{x}; \mathbf{Z}) - f_c(\mathbf{x})]^2$$
$$+ \lambda \frac{1}{|\mathcal{Y}|} \sum_{c \in \mathcal{Y}} \frac{1}{J_c} \sum_{j \in \{1, \cdots, J_c\}} \max_{\mathbf{x}_{jc} \in \mathcal{X}} [\bar{f}_c(\mathbf{x}_{jc}; \mathbf{Z}) - \max_{k \in \mathcal{Y} \backslash c} \bar{f}_k(\mathbf{x}_{jc}; \mathbf{Z})], \quad (4)$$

where $\lambda > 0$. The first term, also used in MM-BD, is the mean squared error between the output of the original model and that of the bounded model. This term is motivated by the hypothesis that the backdoor does not affect the logits of clean (backdoor trigger-free) samples – only samples possessing the backdoor trigger (or a close facsimile of the trigger). In this case, the original network's logits $f_c(\cdot)$ on $\mathcal{D}$ can be treated as *real-valued* supervision, which is much more informative than (discrete-valued) class labels. The second term penalizes the maximum achievable margin of each class (averaged over $J_c$ local maxima). $J_c$ was chosen as the number of clean samples available in each class ($\forall c$, $J_c = |\mathcal{D}|_c = 50$ for CIFAR-10).

To minimize (4) over $\mathbf{Z}$, we perform alternating optimizations. We first initialize $\mathbf{Z}$ conservatively (making the bounds not overly small). We then alternate: 1) gradient ascent to obtain each of the $J_c$ local MMs given $\mathbf{Z}$ held fixed; 2) gradient descent with respect to $\mathbf{Z}$ given the MMs held fixed. These alternating optimizations are performed until a stopping condition is reached (we used the loss difference between two consecutive iterations is less than a threshold $10^{-4}$) or until a pre-specified maximum number of iterations (we use 300 in all the experiments) is reached.

We also suggest an improvement on MMAC that leverages both the original and MMAC-mitigated models. Specifically, this approach detects a backdoor trigger sample (and mitigates by providing a corrected decision on such samples) if the decisions of the two networks differ, or even if their decisions do not differ, if the difference in classification confidence between the two networks on the given sample is anomalous with respect to a null distribution on this difference. Specifically, we used the samples $\mathcal{D}$ to estimate a Gaussian null. A detection is declared when the p-value is less than $\theta = 0.005$, i.e., with a 99.5% confidence. We dub this method extension of MMAC as Maximum-Margin Defense Framework (MMDF).

The benefits of these methods for post-training backdoor mitigation will be demonstrated in our experiments. Also, some analysis supporting our approach is given in the Appendix.

## 3.2 MITIGATING NON-MALICIOUS OVERFITTING: MMOM

Overfitting caused by over-training or data imbalance is different from that caused by backdoor poisoning since there will be a significant generalization performance drop for the former unlike the latter[3]. As shown in Sec. 5.2 below, minimizing (4) is unsuitable for the former. This should in fact be obvious because, to address non-malicious overfitting, one should not *preserve* the logits on clean samples – one can only mitigate such overfitting if these logits are allowed to *change*. So, to mitigate non-malicious overfitting, we minimize the following Maximum-Margin Overfitting Mitigation (MMOM) objective to learn the activation upper bounds:

$$\mathcal{L}(\mathbf{Z}, \lambda; \mathcal{D}) = \frac{1}{|\mathcal{D}|} \sum_{(x,y) \in \mathcal{D}} \mathcal{L}_{CE}(\bar{f}(\mathbf{x}; \mathbf{Z}), y)$$
$$+ \lambda \frac{1}{|\mathcal{Y}|} \sum_{c \in \mathcal{Y}} \frac{1}{J_c} \sum_{j \in \{1, \cdots, J_c\}} \max_{\mathbf{x}_{jc} \in \mathcal{X}} [\bar{f}_c(\mathbf{x}_{jc}; \mathbf{Z}) - \max_{k \in \mathcal{Y} \backslash c} \bar{f}_k(\mathbf{x}_{jc}; \mathbf{Z})], \quad (5)$$

where here $\mathbf{x}$ is a clean input sample, $y \in \mathcal{Y}$ is its class label, and $\mathcal{L}_{CE}$ is the cross-entropy loss.

We minimize Eq. (5) over $\mathbf{Z}$ (again) by two alternating optimization steps:

---

[3]If backdoor poisoning were to degrade generalization accuracy of the model, the model may not be used (it may be discarded). The backdoor attacker of course needs the poisoned model to be used. Thus, a backdoor attack is not really a successful one if it degrades generalization accuracy. Moreover, low poisoning rates are typically sufficient to induce learning of the backdoor trigger while having very modest impact on generalization accuracy of the trained model.

1. Given $\mathbf{Z}$ held fixed, gradient-based margin maximization is used to obtain $\{\mathbf{x}_{jc}; j = 1, \cdots, J_c\}$ for each class $c$:

$$\mathbf{x}_{jc} = \arg\max_{\mathbf{x}_{jc} \in \mathcal{X}} [\bar{f}_c(\mathbf{x}_{jc}; \mathbf{Z}) - \max_{k \in \mathcal{Y}\backslash c} \bar{f}_k(\mathbf{x}_{jc}; \mathbf{Z})], \qquad (6)$$

2. gradient-based minimization of Eq. (5) with respect to $\mathbf{Z}$ given $\{\mathbf{x}_{jc}\}$ held fixed, yielding $\mathbf{Z}^*$ and the mitigated classifier $\bar{f}(\mathbf{x}; \mathbf{Z}^*)$.

## 4 EXPERIMENTS

In this section, we evaluate the performance of our methods on a ResNet-32 model (He et al., 2016) for the CIFAR-10 (10 classes) and CIFAR-100 (100 classes) image datasets (Krizhevsky, 2012). In the original datasets, there are 60000 colored $32 \times 32$-pixel images, with 50000 for training, 10000 for testing, and the same number of samples per category (class-balanced). In our non-malicious overfitting experiments, unbalanced training sets are used, but the test set remains class-balanced.

### 4.1 MITIGATING BACKDOORS

We first consider post-training backdoor mitigation, comparing our MMAC and MMDF against NC (Wang et al., 2019), NAD (Li et al., 2021), FP (Liu et al., 2018a), I-BAU (Zeng et al., 2021), and MM-BM (Wang et al., 2022) defenses (the last also imposes activation clipping). A variety of backdoor attacks are considered including: a subtle, additive global chessboard pattern (Liao et al., 2019); patch-replaced BadNet (Gu et al., 2019); a blended patch (Chen et al., 2017); WaNet (Nguyen & Tran, 2021); an Input-aware attack (Nguyen & Tran, 2020); and a Clean-Label attack (Turner et al., 2018) (where successful Clean-Label attacks .

Results for mitigation of these backdoor attacks are shown in Table 1. As seen, NC, NAD, and FP reduce the attack success rate (ASR) significantly on some attacks, but not others. MMBM reduces ASR substantially on most attacks except for the global chessboard. I-BAU performs well except for the Clean-Label attack. MMAC performs comparably to I-BAU, substantially reducing the ASR for all attacks except for the chessboard pattern. Finally, the MMDF extension of MMAC is the best-performing method, with maximum ASR (across all attacks) of just 3.37%. Moreover, the reduction in clean test accuracy (ACC) relative to the "without defense" baseline is quite modest for MMAC and MMDF (and for I-BAU). By contrast, NAD and NC degrade ACC unacceptably on some of the datasets. Note that the performance of NC mitigation is not applicable (n/a) to not-attacked models because it is predicated on backdoor detection.

### 4.2 MITIGATING NON-MALICIOUS OVERFITTING

Following the design in (Cao et al., 2019; Cui et al., 2019), we downsample the training set to achieve class imbalances. In our work, we consider two types of class priors: exponential (a.k.a. long-tail (LT)) distribution and step distribution. For the LT distribution, the number of samples in each class, $i$, follows the exponential distribution $n_i = n_0 \times \mu_e^i$, while for step distribution, the number of samples in each class follows the step function $n_i = n_0, i < C/2$, and $n_i = \mu_s \times n_0, i \geq C/2$. Here, $\mu_e$ and $\mu_s$ are constants that are set to satisfy different imbalance ratios $\gamma$, which is the sample size ratio between the most frequent class and the least frequent class. We consider two imbalance ratios: 100 and 10. In our experiments, we identify the imbalance type by the distribution name and the imbalance ratio; e.g., "LT-100" means we used the long-tail distribution and chose the imbalance ratio as $\gamma = 100$.

In our experiments, the model trained with standard cross-entropy loss (CE) is used as a baseline. The mixup (Zhang et al., 2017) and GCL (Li et al., 2022) methods are compared with our method. Mixup is a data-augmentation method that creates new training samples by mixing two images and their one-hot labels. In this way, one can increase the number of training samples associated with rare categories, thus achieving a more balanced training set. GCL adjusts the class logits during training, informed by the number of training samples in each class. It should be emphasized that MMOM is designed for the post-training scenario, with no access to the training set and no control of nor knowledge of the training process. By contrast, mixup has access to (and contributes samples to) the training set and knowledge of the degree of class imbalance. Likewise, GCL exploits knowledge

| defense | $|\mathcal{D}|$ | | chessboard | BadNet | blend | WaNet | Input-aware | Clean-Label | no attack |
|---|---|---|---|---|---|---|---|---|---|
| without defense | 0 | PACC | 0.59±0.84 | 0.57±0.76 | 4.67±7.45 | 3.48±1.83 | 5.50±3.30 | 5.72±5.36 | n/a |
| | | ASR | 99.26±1.14 | 99.41±0.76 | 95.08±7.79 | 96.25±1.94 | 94.26±3.45 | 94.13±5.56 | n/a |
| | | ACC | 91.26±0.35 | 91.63±0.35 | 91.57±0.39 | 90.94±0.30 | 90.65±0.34 | 88.68±1.26 | 91.41±0.16 |
| NC | 500 | PACC | 31.98±24.44 | 62.03±17.80 | 25.84±33.09 | 0.30±0.38 | 11.24±9.38 | 83.85±1.81 | n/a |
| | | ASR | 63.72±27.20 | 28.13±19.43 | 71.01±36.94 | 99.64±0.45 | 86.10±11.73 | 20.72±13.84 | n/a |
| | | ACC | 88.49±1.66 | 87.86±1.70 | 87.10±2.59 | 72.25±10.60 | 81.81±1.25 | 68.37±10.04 | 91.41±0.16 |
| NAD | 200 | PACC | 79.88±1.92 | 81.31±3.47 | 86.43±0.63 | 65.33±12.62 | 72.07±4.24 | 76.11±5.83 | n/a |
| | | ASR | 2.92±1.56 | 5.52±2.94 | 1.92±0.70 | 18.03±14.46 | 5.96±7.37 | 10.54±9.17 | n/a |
| | | ACC | 81.28±1.75 | 86.52±1.53 | 87.07±1.30 | 82.39±8.81 | 79.92±4.02 | 82.76±1.39 | 88.87±0.51 |
| FP | 500 | PACC | 19.25±16.53 | 10.91±14.41 | 14.47±13.19 | 83.03±6.40 | 86.04±1.96 | 74.60±8.91 | n/a |
| | | ASR | 55.53±37.34 | 86.49±17.35 | 84.44±14.02 | 0.37±0.23 | 4.09±2.71 | 13.81±11.50 | n/a |
| | | ACC | 90.73±0.60 | 91.83±0.77 | 92.78±0.66 | 90.71±0.78 | 90.71±1.17 | 84.57±1.42 | 90.61±0.77 |
| I-BAU | 100 | PACC | 84.59±3.97 | 85.49±3.32 | 82.45±8.53 | 81.95±5.63 | 80.59±4.27 | 56.95±18.51 | n/q |
| | | ASR | 3.16±3.10 | 2.45±3.74 | 6.61±8.78 | 4.62±5.64 | 1.62±1.06 | 35.59±21.93 | n/a |
| | | ACC | 89.62±0.45 | 89.74±0.57 | 87.87±1.11 | 87.93±2.40 | 89.37±0.71 | 86.69±1.85 | 89.30±0.43 |
| MM-BM | 50 | PACC | 14.19±21.55 | 86.53±1.19 | 86.92±3.96 | 78.23±9.35 | 79.75±1.47 | 85.79±1.56 | n/a |
| | | ASR | 81.77±25.58 | 1.49±0.87 | 3.45±1.73 | 9.85±11.33 | 1.57±1.16 | 1.79±0.9 | n/a |
| | | ACC | 88.48±0.61 | 89.12±0.77 | 88.67±0.36 | 86.24±9.36 | 86.08±1.41 | 86.16±1.10 | 88.70±0.44 |
| MMAC (ours) | 50 | PACC | 52.12±32.08 | 83.57±3.06 | 85.18±2.64 | 84.20±1.75 | 81.86±2.23 | 85.76±1.26 | n/a |
| | | ASR | 26.19±36.49 | 2.21±1.84 | 4.00±2.51 | 4.24±2.32 | 1.70±1.04 | 1.74±0.99 | n/a |
| | | ACC | 87.92 ±0.97 | 88.42±0.79 | 88.23±0.68 | 88.00±0.95 | 88.98±0.64 | 86.21±1.16 | 87.47±0.94 |
| MMDF (ours) | 50 | PACC | 67.35±15.72 | 86.01±1.50 | 86.92±1.63 | 85.02±2.37 | 80.81±2.53 | 86.81±1.63 | n/a |
| | | ASR | 0.45±0.64 | 0.09±0.14 | 1.24±1.18 | 2.66±1.31 | 3.37±2.40 | 0.01±0.01 | n/a |
| | | ACC | 89.12±0.44 | 89.42±0.41 | 90.04±0.40 | 87.02±0.65 | 89.22±0.44 | 86.79±1.41 | 88.99±0.45 |

Table 1: Performance of different mitigation methods under various backdoor data-poisoning attacks on a CIFAR-10 classifier. ASR is the success rate in misclassifying test samples with the backdoor trigger to the target class. ACC is the accuracy on clean test samples. PACC is the accuracy of a mitigated classifier in classifying test samples with the backdoor trigger.

| | CIFAR-10 | | | | CIFAR-100 | | | |
|---|---|---|---|---|---|---|---|---|
| imbalance type | LT-100 | LT-10 | step-100 | step-10 | LT-100 | LT-10 | step-100 | step-10 |
| CE (baseline) | 73.36 | 87.80 | 67.86 | 86.12 | 40.91 | 59.16 | 39.18 | 53.08 |
| mixup (Zhang et al., 2017) | 74.92 | 88.19 | 66.35 | 86.16 | 41.720 | 59.24 | 39.19 | 53.62 |
| GCL (Li et al., 2022) | 82.51 | 89.59 | 81.85 | 88.63 | 46.85 | 60.34 | 46.50 | 58.76 |
| MMOM (ours) | 80.98 | 88.43 | 77.37 | 88.21 | 41.42 | 59.60 | 37.08 | 54.46 |
| MMOM + GCL (ours) | 82.04 | 89.45 | 81.71 | 88.41 | 46.34 | 59.88 | 44.21 | 57.45 |

Table 2: Test set accuracy (%) of different methods on CIFAR-10 and CIFAR-100.

of the class imbalances while also *controlling* the training process. That is, the comparison between MMOM and these two mitigation methods is somewhat unfair to MMOM. For MMOM we do assume that a small balanced dataset $\mathcal{D}$ is available: for CIFAR-10, we take 50 images per class and for CIFAR-100 we take 5 images per class to form this small set. Also, since MMOM is applied after the DNN classifier is trained (while other methods modify the training data or the training process), it is easily combined with these other methods. We apply MMOM both to the baseline model and to the model trained with GCL.

The test set performance is shown in Tab. 2. MMOM improves the test accuracy relative to the CE baseline under all cases except for the "step-100" imbalance type for the CIFAR-100 dataset. For this case, for the 50 rare classes, there are only 5 images per class. So, the class-discriminative features of the rare classes are not adequately learned by the pre-trained model (accuracies for most of the rare classes are zero). Thus, MMOM, which does not modify any DNN model parameters, cannot make any improvement on the rare classes; on the other hand, the accuracy on the common classes will drop due to the activation clipping. This explains the performance drop on the CIFAR-100 dataset with the "step-100" imbalance type. The per-class classification accuracies of MMOM on CIFAR-10 are shown in Fig. 1. MMOM significantly improves the classification accuracies of the rare classes, with only moderate drops in accuracy on the common classes. However, for a model trained with GCL, for which the biased prediction is already mitigated (as shown in (Li et al., 2022)), MMOM fails to improve test accuracy but does not significantly (less than 1%) reduce the accuracy.

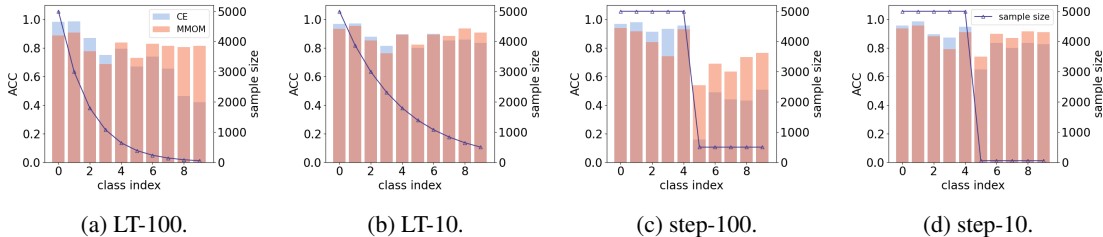

(a) LT-100.    (b) LT-10.    (c) step-100.    (d) step-10.

Figure 1: Post-training results on CIFAR-10. The curves represent the number of samples (sample size) of each class and the bars represent the test set accuracy (ACC) of each class after baseline cross-entropy training and then applying our post-training MMOM method.

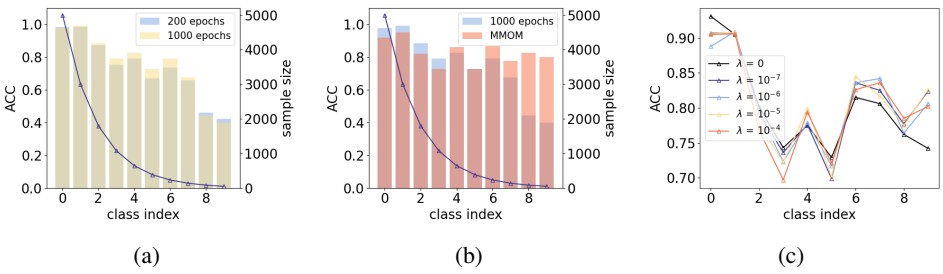

(a)        (b)        (c)

Figure 2: The effect of over-training: (a) The per-class test set accuracies for the model trained for 200 and 1000 epochs. (b) The per-class accuracies after applying our MMOM method. (c) The per-class accuracies under different hyper-parameter ($\lambda$) values. All these results are for CIFAR-10 with imbalance type "LT-100".

### 4.3 MITIGATING WHEN THERE IS OVER-TRAINING

In our main experiments, the models are trained for 200 epochs. Here we evaluate the effect of over-training on the unbalanced CIFAR-10 dataset with type "LT-100", training the model for 1000 epochs. With more training epochs, the overall test accuracy increased from 73.43% to 75.14%, but the impact of class imbalance is amplified – as shown in Fig. 2(a), the accuracies of common classes increase but the accuracies of the two most rare classes decrease. As shown in Fig. 2(b), the impact of this over-training is mitigated by MMOM. Moreover, the overall accuracy increased to 82.84%. Notably, this is better than the accuracy of MMOM applied to the nominal model trained for 200 epochs (80.98% as shown in Tab. 2).

### 4.4 COMPUTATIONAL COMPARISONS

All experiments in our paper were conducted on an NVIDIA RTX-3090 GPU. The computational costs of CE, mixup, and GCL respectively are 1517.52s, 1665.23s, and 1850.84s for the ResNet-32 model on the CIFAR-10 dataset. Under the same setting, the (post-training) computational costs of MMAC and MMOM respectively are 261.46s and 156.67s. Owing to the minimax optimization, MMAC and MMOM computation is significantly longer than the other backdoor mitigation approaches of Tab. 1: Tab. 4 of (Wang et al., 2022) reports times between 20-50s for them for a ResNet-18 model on CIFAR-10 using a similar GPU.

## 5 ABLATION STUDY AND ADDITIONAL EXPERIMENTS

### 5.1 THE HYPER-PARAMETER

In MMOM's minimization objective Eq. (5), $\lambda$ is a positive hyperparameter controlling the weights on the two terms in the objective function. We now consider the performance of MMOM under different values of this hyper-parameter. This is given in Tab. 3, which shows that there is a significant range of $\lambda$ values (from 0 to $10^{-4}$) that give similar performance (with the performance worst at

| $\lambda$ | 0 | $10^{-7}$ | $10^{-6}$ | $10^{-5}$ | $10^{-4}$ | $10^{-3}$ |
|---|---|---|---|---|---|---|
| ACC (%) | 80.09 | 80.80 | 80.57 | 80.77 | 80.41 | 72.50 |

Table 3: Evaluation accuracy over all the classes for different values of $\lambda$ on CIFAR-10.

| | CIFAR-10 | | | | CIFAR-100 | | | |
|---|---|---|---|---|---|---|---|---|
| imbalance type | LT-100 | LT-10 | step-100 | step-10 | LT-100 | LT-10 | step-100 | step-10 |
| CE (baseline) | 73.36 | 87.80 | 67.86 | 86.12 | 40.91 | 59.16 | 39.18 | 53.08 |
| MMAC - Eq. (4) | 72.63 | 87.78 | 67.62 | 86.13 | 40.39 | 58.87 | 39.08 | 52.70 |

Table 4: Classification accuracy using Eq. (4), i.e., MMAC, in trying to mitigate data imbalance.

| | CIFAR-10 | CIFAR-100 |
|---|---|---|
| CE | 93.20 | 69.64 |
| MMOM | 93.08 | 69.41 |

Table 5: Performance on balanced data without over-training.

$\lambda = 0$). That is, our method is not very sensitive to the choice of $\lambda$. Also, the per-class accuracies in Fig. 2(c) show that $\lambda = 0$ gives higher accuracy on the commonest class (class 0) and lower accuracy on the rarest class (class 9). This indicates that the second (MM) term in our MMOM objective plays an important role in eliminating bias in the predicted class.

## 5.2 DIFFERENT ACTIVATION CLIPPING OBJECTIVES

In this subsection, we discuss our two different objective functions for activation clipping. MMOM uses Eq. (5) as the objective to learn the ReLU activation upper bounds. On the other hand, MMAC minimizes Eq. (4) to mitigate overfitting caused by backdoor poisoning. Here we report the performance, when minimizing Eq. (4), on overfitting (biases) of a model trained on unbalanced data. Tab. 4 shows that MMAC does not improve and in fact may harm the performance of a pre-trained model with bias which is not caused by malicious backdoor data-poisoning. This is not surprising – as discussed earlier, MMAC tries to *preserve* the class logits on $\mathcal{D}$, whereas these logits must be changed in order to mitigate non-malicious overfitting.

## 5.3 PERFORMANCE OF MMOM ON A BALANCED DATASET

Under the post-training scenario, only a pre-trained model and a small clean dataset are available. It is not known if the model was trained on an unbalanced dataset or if there is overtraining. Tab. 5 shows the result of applying our MMOM method on a model trained (but not over-trained) on balanced data (the original CIFAR-10 and CIFAR-100). The results show that our method does not degrade the performance of a good model.

## 6 CONCLUSIONS

In this work, we proposed to use MM-based activation clipping to *post-training* mitigate both malicious overfitting due to backdoors and non-malicious overfitting due to class imbalances and overtraining. Whereas maximizing margin is a very sensible objective for support vector machines, it would lead to gross overfitting (too-confident predictions) for DNN models, which are vastly paremterized compared to SVMs. Thus, penalizing (limiting) MMs (the opposite of margin maximization) is most suitable to mitigate overfitting in DNNs. Our MMOM method, designed to mitigate nonmalicious overfitting post-training, does not require any knowledge of the training dataset or of the training process and relies only on a small balanced dataset. Results show that this method improves generalization accuracy both for rare classes as well as overall. Our related MMAC method and its extension, MMDF, designed to post-training mitigate backdoors, are agnostic to the manner of backdoor incorporation, i.e. they aspire to achieve universal backdoor mitigation. These methods were

shown to outperform a variety of peer methods, for a variety of backdoor attacks. MMOM, MMAC, and MMDF are computationally inexpensive. This paper demonstrates an approach (MMOM) originating from (motivated by) adversarial learning which, unlike adversarial training (Madry et al., 2018), *helps* clean generalization accuracy, rather than harming it. Future work may investigate whether MM is also effective at mitigating overfitting when used as a regularizer *during training*. We may also consider applying MMDF and MMOM *in parallel*, as in practice (in the absence of performing backdoor detection) we may not know whether potential overfitting has a malicious or a non-malicious cause. Future work may also investigate other ideas/approaches from adversarial learning that may provide further benefits in helping to achieve robust deep learning, i.e. good generalization accuracy.

## A   APPENDIX: SOME ANALYSIS SUPPORTING MMAC

Denote the (positive) class-$s$ logit as $\phi_s$ before activation clipping and $\bar{\phi}_s$ after. Let $\bar{f}_{t-s} = \bar{\phi}_t - \bar{\phi}_s$. Suppose a small additive backdoor pattern $\delta$ with source class $s$ and target class $t \neq s$. Consider an arbitrary sample $x_s$ following the nominal class-$s$ distribution. To a first-order approximation in $\|\delta\|$, by Taylor's theorem,

$$
\begin{aligned}
\bar{f}_{t-s}(x_s + \delta) &\approx \bar{\phi}_t(x_s) + \bar{\phi}(x_s) + \delta^T(\nabla\bar{\phi}_t(x_s) - \nabla\bar{\phi}_s(x_s)) \\
&\leq \bar{\phi}_t(x_s) + \bar{\phi}(x_s) + \|\delta\| \cdot \|(\nabla\bar{\phi}_t(x_s)\| - \delta^T\nabla\bar{\phi}_s(x_s),
\end{aligned}
\tag{7}
$$

where the inequality comes from applying Cauchy-Schwarz.

One can directly show (Wang et al., 2022) via a first-order Taylor series approximation that if the classification margins on the training dataset are lower bounded after deep learning so that there is no class confusion, i.e., $f_{s-t}(x_s) > \tau > 0$ and $f_{t-s}(x_s + \delta) > \tau$ (the latter a backdoor poisoned sample), then the inner product (directional derivative)

$$
\delta^T\nabla f_{t-s}(x_s) > 2\tau.
\tag{8}
$$

This inequality indicates how deep learning overfits backdoor pattern $\delta$ to target class $t$ so that the presence of $\delta$ "overcomes" the activations from the nominal class-$s$ discriminative features of $x_s$.

Under the proposed activation clipping, logits for any class $i$ are *preserved* for data following the class-$i$ distribution; thus,

$$
\phi_s(x_s) \approx \bar{\phi}_s(x_s) \text{ and } \nabla\phi_s(x_s) \approx \nabla\bar{\phi}_s(x_s).
$$

Clipping an ReLU bounds its output and increases the domain over which its derivative is zero. By reducing the unconstrained-maximum classification margin, logits for any class $i$ may be reduced for samples which do **not** follow the nominal class-$i$ distribution. In this case, $\phi_t(x_s) > \bar{\phi}_t(x_s)$ and $\|\nabla\phi_s(x_s)\| > \|\nabla\bar{\phi}_s(x_s)\|$. That is, activation clipping may reduce the norm of the logit gradients, compared to their nominal distribution. Although $f_{t-s}(x_s + \delta) > 0$ for the original poisoned DNN, the previous two inequalities may be significant enough so that the right-hand-side of Eq. (7) is negative, i.e. so that a backdoor trigger does not cause the class to change from $s$ to $t$ in the activation-clipped DNN. Finally note that the "overfitting inequality" (8) may not hold for the activation-clipped DNN.

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
