# OpenReview forum: "Maximum Margin Based Activation Clipping for Post-Training Overfitting Mitigation in DNN Classifiers"
_ICLR.cc/2024/Conference — ICLR 2024 Conference Withdrawn Submission_

### Official Review · Reviewer_EQmf · 2023-10-27

**Soundness:** 3 good
**Presentation:** 3 good
**Contribution:** 2 fair
**Rating:** 5
**Confidence:** 3

**Summary:**

The researchers introduced a method to reduce overfitting in deep neural networks. They focused on addressing both intentional and unintentional overfitting issues. Their main approach, called MMOM, improves accuracy without needing extensive data or training details. They also developed methods, MMAC and MMDF, to tackle hidden malicious changes (backdoors) in models. These methods were found to be effective and efficient compared to others. The study emphasized that their techniques, inspired by adversarial learning, actually improve model accuracy. Future research might explore using these methods during training or delve deeper into adversarial learning to further enhance DNN performance.

**Strengths:**

1. This paper introduces a novel method using activation clipping as a means to counteract both backdoor attacks and overfitting challenges.

2. To validate the efficacy of their proposed technique, the authors have carried out a comprehensive set of experiments.

**Weaknesses:**

1. I understand that the big topics is the overfitting problem. However, it is wired to choose the two topics: backdoor attacks and imbalanced dataset for evaluating this problem. For example, L-2 regularization, or dropout are proposed to overcome the overfitting problem, but they will not test the regularization effect on these two topics.
2. The authors assert that their proposed method prevents overfitting. However, the experimental results appear to indicate that, upon implementing activation clipping, the model's clean accuracy actually deteriorates. Generally, one would anticipate that addressing overfitting might enhance model performance, but this proposed method doesn't seem to achieve that.

**Questions:**

Please refer to the weaknesses.

---

> ### Author Response · Authors · 2023-11-16
> **Response to Reviewer EQmf:**
>
> Thank you for your encouraging feedback, and our response to the comments are as follows:
>
>
>
>
> We agree with the reviewer that this connection between backdoors and non-malicious bias (long-tailed issues) is surprising, which leads to the novel idea (which we advocate in this paper) that a variation of a backdoor defense can be used to mitigate non-malicious bias post-training. L-2 regularization and dropout could be investigated for both problems.
>
>
> We think the reviewer is referring to Table 5 reporting results applying MMOM to models that were not over-trained on training datasets that were class-balanced, i.e., “clean” scenario. The reviewer’s comment is confusing to us because our reported results show just 0.2% drop in overall accuracy. GCL is a during-training method which assumes knowledge of the number of samples per class. So, with balanced data, GCL is just normal training for the “clean” scenario. We think that the modest 0.2% drop is acceptable for a post-training method (like MMOM).

---

### Official Review · Reviewer_qM5Q · 2023-10-29

**Soundness:** 2 fair
**Presentation:** 3 good
**Contribution:** 2 fair
**Rating:** 3
**Confidence:** 4

**Summary:**

This paper proposes an activation clipping method, which can be used to remove backdoors and improve model generalizability. The paper hypothesizes that backdoor-injected inputs lead to large activation values compared to clean inputs. It hence proposes to limit the upper bound of activation values for every layer in the model. Specifically, it optimizes a set of bounding vectors that are used to constrain the activation values. This method is also applicable to improving the model's generalization under the imbalanced data setting. The evaluation is conducted on a ResNet model and two image datasets. The results show that the proposed approach is effective in removing backdoors compared to a few baselines. It also demonstrates reasonable performance in improving generalization accuracy.

**Strengths:**

1. The paper aims to address an important and timely problem: backdoor attacks pose a security threat to deep learning models. The proposed approach in this paper can be used to mitigate backdoors in trained models. Additionally, this method is applicable to improving models' generalization on imbalanced data, resulting in higher test accuracy.

2. The paper is mostly clear. The description of the main technique is easy to understand.

**Weaknesses:**

1. The proposed method addresses a narrow subject, applicable only to models with the ReLU activation function. While ReLU is widely used, it is limiting to have an approach tailored exclusively for it. It would be beneficial to show the performance of the proposed technique with other activation functions as well.

2. The assumption made in the paper is that backdoor attacks will yield larger activation values compared to clean inputs. Are there any empirical evidences supporting this assumption? This paper evaluates several attacks in the experiments. However, there are many other advanced attacks, such as filter attack, reflection attack [1], DFST [2], dynamic attack [3], composite attack [4], and SIG [5]. Do all of these exhibit large activation values compared to clean inputs? It would be more convincing to have a study to show the generalization of the proposed method.

3. The paper only evaluates known attacks. As the proposed approach relies on certain assumptions, an adaptive attack could intentionally violate these assumptions. For instance, an attacker might manipulate the activation values of a backdoor-injected input to closely resemble the activation value ranges of clean inputs. Would the proposed technique still be effective in such a scenario?"

4. In the experiment on mitigating non-malicious overfitting, which data samples are used for evaluation? It appears that they are not from the training set. This raises a concern about the fairness of comparison with other baselines. The paper argues that it is unfair to evaluate the proposed approach since it does not have access to the training set. However, given that it is a post-training technique, if it utilizes balanced non-training data, this implies an advantage over the baselines. All of those baselines only have access to imbalanced data.

5. Only two baselines are compared in Section 4.2. I am not an expert on the long-tail problem, but by simply searching online, I found a large number of works aiming to address it. Are the two baselines compared in the paper representative? Why were other related techniques not compared?

6. The evaluation conducted in this study is limited. Only one model was used in the experiment, which hinders the assessment of the technique's generalizability. The results reported in Table 1 may be overfitting on this particular model. There are advanced model architectures that have been shown to be more powerful than ResNet, such as vision transformers. Does the proposed technique work on these new architectures? Furthermore, it would be better to conduct experiments on large image datasets, such as ImageNet.


[1] Liu, Y., Ma, X., Bailey, J., and Lu, F. Reflection backdoor: A natural backdoor attack on deep neural networks. ECCV 2020.\
[2] Cheng, S., Liu, Y., Ma, S., and Zhang, X. Deep feature space trojan attack of neural networks by controlled detoxification. AAAI 2021.\
[3] Salem, A., Wen, R., Backes, M., Ma, S., and Zhang, Y. Dynamic backdoor attacks against machine learning models. EuroS&P 2022.\
[4] Lin, J., Xu, L., Liu, Y., and Zhang, X. Composite backdoor attack for deep neural network by mixing existing benign features. CCS 2020.\
[5] Barni, M., Kallas, K., & Tondi, B. (2019, September). A new backdoor attack in cnns by training set corruption without label poisoning. ICIP 2019.

**Questions:**

See weaknesses section.

---

> ### Author Response · Authors · 2023-11-16
> **Response to Reviewer qM5Q**
>
> We sincerely thank you for the time taken to provide detailed reviews which will improve our paper. Our response to the comments are as follows:
>
> 1. ReLUs are widely used – in fact, many papers don’t even mention the activation function they use because it is so commonplace to use ReLUs. We already have experimental results showing that our method works with other types of neural-activation functions and we will include them in an appendix of the revision.
>
> 2. We will also show experimental results for our backdoor mitigation against the attacks this reviewer notes (as for the previous reviewer). Again, note that SIG is global, additive backdoor patterns like the subtle chessboard we’ve already evaluated. We have previously plotted histograms of neural activations which demonstrate that localized backdoor patterns do result in high activations but global patterns (like chessboard) may not (both result in anomalously large classification margins). This was a main motivation behind improving the MM-BM method to MMAC, though, again, we acknowledge that the two methods are similar.
>
> 3. We have already evaluated adaptive attacks (as in the MM-BD paper) for MMAC but neglected to include them in an appendix of this submission. In an appendix of the revised paper, for MMAC we will include the adaptive attack results, models with leaky ReLUs, models trained on other datasets.
>
> 4. Other baselines are during training not post-training. In the revision, we can consider MMOM based on a commensurately large, class-balanced subset (randomly sampled) from the training dataset. We expect that the performance results for MMOM will not degrade by using training samples. Similarly, if the Kang et al. ’20 method instead employs non-training data that is similarly distributed (i.e., make it a post-training method) we expect that its performance will not improve. This said, we will compare against both of these modified methods (post-training Kang and a during-training MMOM).
>
> 5. We will compare against additional methods which try to address the long-tailed problem (including two called out by a previous reviewer), but note that this reviewer does not indicate any specific prior works, particularly any that operate post-training like our MMOM. We also note that during-training GCL, which performs comparably with post-training MMOM, is demonstrated to be superior to seven other methods in the GCL paper using the same experimental scenario that we do.
>
> 6. We have already done some of these experiments and will provide expanded experimental results in an appendix of the revised paper to satisfy this request.

---

### Official Review · Reviewer_Ww2o · 2023-11-05

**Soundness:** 2 fair
**Presentation:** 1 poor
**Contribution:** 2 fair
**Rating:** 3
**Confidence:** 4

**Summary:**

This paper proposes a novel maximum margin objective to mitigate two different types of over-fitting that happens in neural networks: one in backdoor attacks and the other in long-tailed classification. Motivated by an earlier work from Wang et al. [1], this paper argues that since the root cause of these two issues lies in neural networks' over-fitting, one might use a unified solution using maximum margin neural networks to mitigate them. To this end, the paper proposes two novel objective functions, named Maximum Margin Activation Clipping (MMAC) and Maximum Margin Over-fitting Mitigation (MMOM), for combating backdoor attacks and long-tailed recognition, respectively. Both objectives aim to find a set of boundary vectors to adjust the internal activation layers of ReLU-based neural networks, effectively preventing them from over-fitting. These objectives are formed as a minmax optimization problem. Using a clean held-out validation set, a solution is found post-training to address backdoor attacks and imbalanced classification. Experimental results on CIFAR-10 and CIFAR-100 demonstrate the effectiveness of these two proposed objectives in combating overfitting related to triggers and imbalanced data.

[1] Wang, Hang, et al. "MM-BD: Post-Training Detection of Backdoor Attacks with Arbitrary Backdoor Pattern Types Using a Maximum Margin Statistic."_IEEE Symposium on Security and Privacy_ (SP), 2024.

**Strengths:**

- The observations made in the paper around a common root-cause for backdoor attacks and long-tailed classification seems novel and interesting. To my best knowledge, no other prior work has looked at these two issues jointly, and this paper might be the first one in this regard.

- The proposes solution is simple yet effective. Since it is applied to the neural networks post-training, one could easily refine their already trained neural networks to address their issues.

- Experiments conducted over CIFAR-10 and CIFAR-100 datasets indicate that the proposed solution could effectively combat backdoor attacks and deliver a more balanced performance on long-tailed datasets. The proposed approach only requires a small subset of clean data and is applicable to models post-training.

**Weaknesses:**

There major issues with this submission can be categorized into three streams:

   1. **Novelty and Applicability**: This paper is a simple extension of an earlier work by Wang et al. [1], and it has limited novelty. Most of the notions used in this paper already appear in the prior work, and this paper reorganizes their objective function and introduces a different regularizer than the one used in [1]. One can also see from the results in Table 1 and also Table 3 that introducing this new regularizer has somehow a minimal effect on the results on average (see Table 3 for how even the smallest values for $\lambda$ could yield similar results to the case where this term is absent from the objective.) Thus, it is unclear whether the changes introduced in this paper are significant or not. _More importantly_, the objectives introduced for MMAC and MMOM assume that the model deployer knows beforehand whether the model is backdoored or suffers from long-tailed issues. Even though detecting whether a model suffers from issues related to imbalanced datasets is easy, but we don't know whether the model has been backdoored or not. As such, one might need to also apply MMAC on top of clean models, which could hurt the benign performance.

   2. **Experimental Settings and Results**: The other significant issue with the current submission is its lack of comprehensive experiments. Many newer attacks, such as SIG [2], ISSBA [3], Refool [4], DFST [5], and WB [6], are not considered for benchmarking. Moreover, there are better baselines, such as those in [7-8], for long-tailed recognition. In particular, contrary to the paper claims that "_To the best of our knowledge, ours is the first work to address class imbalances post-training,_" I firmly believe that the seminal work of Kang et al. 2020 is the first work around this idea and as such, it should be both cited and used as a baseline. This is since Kang et al. 2020 just fine-tune the classification layer of the classifiers and as such, do not require re-training on the entire imbalanced data. Finally, most works in backdoor attacks and specially in long-tailed recognition also report their results over the ImageNet and ImageNet-LT datasets which this paper also needs to include these results.

   3. **Writing, Structure, and Presentation**: Last but not least, this paper requires major changes in its narrative and formatting. _First_, instead of giving a standard overview of the work by highlighting the contributions in the introduction, the paper shocks the reader and goes into the related work section before pointing out its contributions. In fact, the majority of the last two paragraphs in the related work section belong to the introduction. _Second_, the paper heavily relies on the reader having read the work of Wang et al. [1], and it uses the information in that paper as presumed knowledge. For instance, instead of giving the intuitions behind Eq. (2) and discussing how it relates to regulation of decision boundaries or over-fitting prevention, the paper just presents the formulation without adequately explaining it. The same issue can be observed around the introduction of $J\_{c}$ in the second sentence of Section 3.1 where the readers might wonder how this is related to the previously discussed notions, and how we ended up with this definition. Finally, the paper jumps around to unrelated topics such as Federated Learning in the introduction, Adversarial Training in the Abstract and the Conclusion, etc. and makes the readers confused. I believe that the discussions around these two notions are not helping with building the correct narrative for the contributions of the paper, and removing them would result in less confusion.

[1] Wang et al. "MM-BD: Post-Training Detection of Backdoor Attacks with Arbitrary Backdoor Pattern Types Using a Maximum Margin Statistic."_IEEE Symposium on Security and Privacy_ (SP), 2024.

[2] Barni et al. "A new backdoor attack in CNNs by training set corruption without label poisoning." _ICIP_, 2019.

[3] Li et al. "Invisible backdoor attack with sample-specific triggers." _ICCV_, 2021.

[4] Liu et al. "Reflection backdoor: A natural backdoor attack on deep neural networks." _ECCV_, 2020.

[5] Cheng et al. "Deep feature space trojan attack of neural networks by controlled detoxification." _AAAI_, 2021.

[6] Doan et al. "Backdoor attack with imperceptible input and latent modification." _NeurIPS_, 2021.

[7] Kang et al. "Decoupling representation and classifier for long-tailed recognition." _ICLR_, 2020.

[8] Yang et al. "Rethinking the value of labels for improving class-imbalanced learning." _NeurIPS_, 2020.

[9] Kirichenko et al. "Last layer re-training is sufficient for robustness to spurious correlations." _ICLR_, 2023.

**Questions:**

Apart from the questions raised above, here are some additional questions/suggestions:

- Can you explain why the discussions around adversarial training is related to the present work? Because adversarial training and robustness to backdoor attacks are two different topics, and nobody in the backdoor literature would accept a degradation in benign accuracy even though this might be acceptable in adversarial robustness.

- Why 200 epochs is used for training the models? Usually just 120 epochs are enough for training ResNet models over small scale datasets such as CIFAR-10 and CIFAR-100.

- In the experiments of Table 3, the data is clean or backdoored? Have you repeated the same experiment for backdoored models?

- Please add the table captions to the top of the tables, not the bottom.

---

> ### Author Response · Authors · 2023-11-16
> **Response to Reviewer Ww2o**
>
> We thank you for your time and effort in reviewing our paper, our responses are as follows:
>
> Response to the weakness:
>
> 1. About the similarity with the MM-BD method, please see our previous response re. comparing this paper with MM-BM (MM-BD [1]). We in fact do not assume a priori that the model suffers from long-tailed issues or backdoors. Both MMAC and MMOM are also tested on clean models. In the revised paper’s appendix, we will add experimental results applying both mitigation methods to four different scenarios: clean models, backdoor poisoned models and no long-tailed issues, models with long-tailed issues which are not backdoor poisoned, and models which are both backdoor poisoned and have long-tailed issues.
>
> 2. Also, We will expand our backdoor experiments to include the attacks which the reviewer notes, but we note, e.g., SIG is additive global backdoor pattern like the subtle chessboard we’ve reported. We do compare against two during-training defenses against long-tailed issues, GCL and Mixup, in the experimental scenario of the GCL paper. The GCL paper demonstrates how GCL compares favorably against seven other methods including Kang et al. 2020. We will expand our evaluation to cite and include comparisons against Kang et al. 2020 and Yang et al. 2020 (cited as [7],[8] below).
>
> 3. Finally, We will accordingly revise the paper on writing, structure and presentation.
>
> Response to the questions:
>
> 1. It's not clear to us why clean accuracy degradation should be acceptable for adversarial robustness but not for backdoor robustness – both try to achieve robustness to adversarial inputs (including backdoor triggers). This said, we agree with the reviewer that this is a digression and we will remove the discussion of adversarial training.
>
>
> 2. 200 epochs of training were used in the GCL paper for ResNet-32.
>
>
> 3. In table 3 we no longer consider backdoor attacks, instead, all the models considered in table 3 are trained on imbalanced datasets.
>
>
> 4. We will revise the captions of tables accordingly.

---

### Official Review · Reviewer_Xe9i · 2023-11-09

**Soundness:** 2 fair
**Presentation:** 3 good
**Contribution:** 3 good
**Rating:** 3
**Confidence:** 3

**Summary:**

This work proposes an improvement over the backdoor mitigation technique proposed by the MM-BD paper in 2022. The authors propose post-training maximum margin-based activation clipping to handle backdoor trojan attacks and non-malicious overfitting due to class imbalances and overtraining. Two proposed methods - MMAC and MMDF, are computationally inexpensive and outperform existing methods for backdoor attacks on the CIFAR10 dataset. In contrast, the third variant, MMOM, enhances test accuracy for rare classes (and overall) for class imbalance and overtraining scenarios on the CIFAR10 and CIFAR100 datasets.

**Strengths:**

1. The idea behind maximum-margin-based activation clipping for overtraining and class imbalance scenarios seems intuitive.

2. MMAC and MMDF comprehensively outperform existing backdoor mitigation methods across various backdoor attack patterns for the CIFAR10 dataset. As such, they are agnostic to the manner of backdoor pattern incorporation. However, some of the experimental settings seem inconsistent, as outlined in Weakness #3.

3. The MMOM model compares favorably with GCL for mitigating biased predictions without knowledge of the training process. In contrast, GCL already has access to training samples and the degree of class imbalance during training for post-training mitigation.

4. The MMOM model does not degrade the test class accuracy of a model trained (but not over-trained) with balanced data - showing the formulation is suitable for both detection and mitigation.

**Weaknesses:**

1. The work builds on the MM-BM methodology by Wang et al., but the backdoor mitigation problem seems to be a not-so-important contribution of the original paper; instead, its methodology and experiments are more focused on the backdoor trigger detection problem. As such, the mitigation experiments were only performed for the relatively small CIFAR-10 dataset. This work also provides results for the non-malicious overfitting scenario on CIFAR-100, but the experiments seem insufficient to judge the overall merit of the idea. Experiments on TinyImageNet and/or GTSRB are lacking.

2. The problem of non-malicious overfitting due to class imbalance and overtraining, addressed for the first time post-training in this work, seems somewhat tangential to adversarial and trojan backdoor attacks. As such, it’s unsurprising that existing mitigation approaches don’t design techniques to tackle this issue. More well-grounded arguments would be good to motivate and connect the 2 issues further.

3. In Table 1 of the MM-BD paper, the authors outline the backdoor embedding types (additive, patch replacement, etc.) and the number of source classes (single or multiple) during the backdoor attack for evaluating detection and mitigation performance. No such details are mentioned in the experiments section in 4.1.

4. The summary of the main contributions is delayed until the end of the Related Work section instead of the Introduction - not a major issue, but it makes the overall flow difficult to follow.

5. The MMOM motivation and methodology in Section 3.2 seems a bit too abrupt and brief for proper understanding.

6. The paper assumes substantial domain knowledge of the reader in trojan backdoor attacks and mitigation and familiarity with the MM-BD paper. Many of the terminologies aren’t as well-explained as existing works do, e.g., Neural Cleanse (Wang et al., 2019).

7. [Minor] Table 1 is very hard to read - quite difficult to compare the performance of the proposed methods with existing work as none of the numbers are highlighted. Also, an up or down arrow beside every metric, like ASR or ACC, would have been helpful to understand if higher or lower is better. Mixup paper citation missing in Section 4.2.

**Questions:**

1. Not entirely clear if the activation clipping is, in any way, different from what was already proposed by Wang et al. (2022). This is claimed as a contribution but seems very similar to Section 5.3 in the MM-BD paper. Also, it would be interesting to see if the second term of the loss function in MM-BD (Eq. 11) would have complementary effects with the new loss term proposed here.

2. The experimental settings for mitigation (Table 1) seem a bit inconsistent with existing work. Wang et al. mention using ResNet-18 for CIFAR-10 and VGG-16 for CIFAR-100, while here, ResNet-32 is used for all methods. Were all the methods re-evaluated in that case with this backbone?

3. [Effect of \lambda] - Table 3 shows a steep drop-off in performance for \lambda= 10-3 and doesn’t improve too much as \lambda is increased from 0 to 10-4. Is the second term of the loss function really relevant for classification performance, then? Also, ablations for the effect of \lambda on the MMAC/MMDF objective (Eq 4) are missing. Not entirely obvious if the hyperparameter would have the same effect for backdoor mitigation as for generalization accuracy.

4. It’s not entirely clear why MMOM combined with GCL slightly degrades GCL’s performance. Since GCL already mitigates biased predictions using information from the training set, MMOM+GCL should, at least, preserve it, if not increase it. The drop-off for CIFAR-100 in the step-100 scenario seems substantial, for example.

---

> ### Author Response · Authors · 2023-11-16
> **Response to Reviewer Xe9i:**
>
> Thank you for the time taken to provide detailed feedback to which we respond below:
>
> Response to the weakness:
>
> 1. We have already performed experiments with other datasets and other models wherein our proposed methods compare favorably. We will include them and additional experiments, as suggested, in an appendix.
>
> 2. We appreciate the reviewer acknowledging that we are among the first to look at the non-malicious overfitting (long-tail issue) post-training. A major contribution of this paper is that, while non-malicious overfitting seems tangential to backdoors, they are in fact closely related. Both class imbalances and trojans are sources of model overfitting and hence similar ideas can be applied to mitigate overfitting attributable to EITHER of these causes. The fact that the reviewer thinks these problems are “tangential” means that the link between these problems (demonstrated by our paper) is surprising, and is thus an indication of both the novelty and significance of this work.
>
> 3. We will provide the details missed in our submission and restructure the paper accordingly. We will clarify our jargon and summarize the MM-BD/MM-BM paper’s approach and results.
>
> Response to the questions:
>
> 1. We acknowledge the methods (Wang et al. 2022 and ours) are similar - the difference between MMAC’s objective and MM-BM’s objective (second term) results in the former’s better performance for global, additive backdoor patterns (chessboard in our case). Additionally, the proposed MMAC backdoor mitigation is superior to MM-BM+FP (as reported in the MM-BD paper) and I-BAU. So, MMAC is preferred. This will be more clearly explained in the revision.
>
>
>
>
> 2. We employ ResNet-32 as that was the architecture used in the GCL paper.
>
>
>
> 3. The impact of small values of lambda is indicated in Fig. 2c. This impact depends both on class imbalance in the training dataset (long-tail issues) and intrinsic confusion between classes.
>
>
> 4. GCL is a during-training approach, which accesses and exploits the training set, while MMOM is post-training (with no access to the training set). That is, MMOM is being applied after GCL. Thus, the overfitting phenomenon is significantly affected by GCL before MMOM is applied. We have no reason to expect that MMOM will improve GCL (or vice versa).